# Exploring Jiu-Jitsu Argumentation for Writing Peer Review Rebuttals

**Sukannya Purkayastha**[1]   **Anne Lauscher**[2]   **Iryna Gurevych**[1]
[1] Ubiquitous Knowledge Processing Lab,
Department of Computer Science and Hessian Center for AI (hessian.AI),
Technical University of Darmstadt
[2] Data Science Group, University of Hamburg
www.ukp.tu-darmstadt.de

## Abstract

In many domains of argumentation, people's arguments are driven by so-called *attitude roots*, i.e., underlying beliefs and world views, and their corresponding *attitude themes*. Given the strength of these latent drivers of arguments, recent work in psychology suggests that instead of directly countering surface-level reasoning (e.g., falsifying given premises), one should follow an argumentation style inspired by the Jiu-Jitsu "soft" combat system (Hornsey and Fielding, 2017): first, identify an arguer's attitude roots and themes, and then choose a prototypical rebuttal that is aligned with those drivers instead of invalidating those. In this work, we are the first to explore Jiu-Jitsu argumentation for peer review by proposing the novel task of *attitude and theme-guided rebuttal generation*. To this end, we enrich an existing dataset for discourse structure in peer reviews with attitude roots, attitude themes, and canonical rebuttals. To facilitate this process, we recast established annotation concepts from the domain of peer reviews (e.g., aspects a review sentence is relating to) and train domain-specific models. We then propose strong rebuttal generation strategies, which we benchmark on our novel dataset for the task of end-to-end attitude and theme-guided rebuttal generation and two subtasks.[1]

## 1 Introduction

Peer review, one of the most challenging arenas of argumentation (Fromm et al., 2020), is a crucial element for ensuring high quality in science: authors present their findings in the form of a publication, and their peers argue why it should or should not be added to the accepted knowledge in a field. Often, the reviews are also followed by an additional *rebuttal phase*. Here, the authors have a chance to convince the reviewers to raise their assessment scores with carefully designed counterarguments.

[1]Code at https://github.com/UKPLab/EMNLP2023_jiu_jitsu_argumentation_for_rebuttals

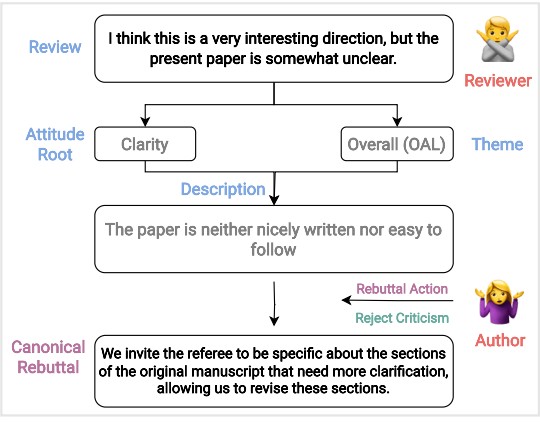

Figure 1: Review to Canonical Rebuttal Mapping via intermediate steps. The review has the attitude root *'Clarity'* and theme *'Overall'*. The canonical rebuttal is for the rebuttal action *'Reject Criticism'*.

Recently, the analysis of review-rebuttal dynamics has received more and more attention in NLP research (Cheng et al., 2020; Bao et al., 2021; Kennard et al., 2022). For instance, previous efforts focused on understanding the links between reviews and rebuttals (Cheng et al., 2020; Bao et al., 2021) and on categorizing different rebuttal actions (e.g., *thanking* the reviewers; Kennard et al., 2022). Gao et al. (2019) find that well-structured rebuttals can indeed induce a positive score change. However, writing good rebuttals is challenging, especially for younger researchers and non-native speakers. For supporting them in writing effective rebuttals, argumentation technology can be leveraged.

In computational argumentation, existing efforts towards generating counterarguments are based on *surface-level reasoning* (Wachsmuth et al., 2018; Alshomary et al., 2021; Alshomary and Wachsmuth, 2023). That is, existing research focuses on directly rebutting the opponents' claims based on what is evident from the argument's surface form. In contrast, researchers also proposed theories that explain how one's arguments are driven by underlying beliefs (Kiesel et al., 2022;

Liscio et al., 2022). One such concept, rooted in psychology, relates to *'attitude roots'* and corresponding finer-grained *'attitude themes'*. These attitude roots are the underlying beliefs that help sustain surface opinions (Hornsey and Fielding, 2017). Based on this idea, the authors further proposed *'Jiu-Jitsu argumentation'*, a rebuttal strategy that relies on understanding and using one's attitude roots to identify generic but customizable counterarguments, termed *canonical rebuttals*. As those are aligned with the opponent's underlying beliefs, Jiu-Jitsu argumentation promises to change opinions more effectively than simple surface-level counterargumentation. In this work, we acknowledge the potential of leveraging the latent drivers of arguments and are the first to explore Jiu-Jitsu argumentation for peer-review rebuttals.

Concretely, we propose the task of attitude root and theme-guided rebuttal generation. In this context, we explore reusing established concepts from peer review analysis for cheaply obtaining reviewer attitude roots and themes (Kennard et al., 2022; Ghosal et al., 2022): reviewing aspects, and reviewing targets (paper sections). We show an example in Figure 1: the example review sentence criticizes the clarity of the paper without being specific about its target. The argument is thus driven by the attitude root *Clarity* and the attitude theme *Overall*. The combination of these two drivers can be mapped to an abstract and generic description of the reviewer's beliefs. Next, given a specific rebuttal action, here *Reject Criticism*, a canonical rebuttal sentence can be retrieved or generated, serving as a template for further rebuttal refinement.

**Contributions.** Our contributions are three-fold: (**1**) we are the first to propose the novel task of attitude root and theme-guided peer review rebuttal generation, inspired by Jiu-Jitsu argumentation. (**2**) Next, we present **JITSUPEER**, an enrichment to an existing collection of peer reviews with attitude roots, attitude themes, and canonical rebuttals. We build JITSUPEER by recasting established concepts from peer review analysis and training a series of models, which we additionally specialize for the peer-reviewing domain via intermediate training. (**3**) Finally, we benchmark a range of strong baselines for end-to-end attitude root and theme-guided peer review rebuttal generation as well as for two related subtasks: generating an abstract review descriptions reflecting the underlying attitude, and scoring the suitability of sentences for serving as

canonical rebuttals. We hope that our efforts will fuel more research on effective rebuttal generation.

## 2 Jiu-Jitsu Argumentation for Peer Review Rebuttals

We provide an introduction to Jiu-Jitsu argumentation and the task we propose.

### 2.1 Background: Jiu-Jitsu Argumentation

'Jiu-Jitsu' describes a close-combat-based fighting system practiced as a form of Japanese martial arts.[2] The term 'Jiu' refers to soft or gentle, whereas 'Jitsu' is related to combat or skill. The idea of Jiu-Jitsu is to use the opponent's strength to combat them rather than using one's own force. This concept serves as an analogy in psychology to describe an argumentation style proposed for persuading anti-scientists, e.g., climate-change skeptics, by understanding their underlying beliefs ('attitude roots', e.g., fears and phobias; Hornsey and Fielding, 2017). The idea is to use one's attitude roots to identify counterarguments which more effectively can change one's opinion, termed *canonical rebuttals*. A canonical rebuttal aligns with the underlying attitude roots and is congenial to those. They serve as general-purpose counterarguments that can be used for any arguments in that particular attitude root–theme cluster.

### 2.2 Attitude Root and Theme-Guided Peer Review Rebuttal Generation

We explore the concept of attitude roots for the domain of peer review. Our goal is to identify the underlying beliefs and opinions reviewers have while judging the standards of scientific papers (cf. §3). Note that a good rebuttal in peer reviewing is also dependent on the specific rebuttal action (Kennard et al., 2022) to be performed like *mitigating criticism* vs. *thanking* the reviewers. For example, in Figure 1, we display the canonical rebuttal for *'Reject Criticism'*. For the same attitude root and theme, the canonical rebuttal for the rebuttal action *'Task Done'* would be *"We have significantly improved the writing, re-done the bibliography and citing, and organized the most important theorems and definitions into a clearer presentation."*. We thus define a canonical rebuttal as follows:

**Definition.** *'A counterargument that is congenial to the underlying attitude roots while addressing*

---

[2]https://en.wikipedia.org/wiki/Jujutsu

*them. It is general enough to serve (as a template) for many instances of the same (attitude root–theme) review tuples while expressing specific rebuttal actions.'*

Given this definition, we propose *attitude root and theme-guided rebuttal generation*: given a peer review argument $rev$ and a rebuttal action $a$ the task is to generate the canonical rebuttal $c$ based on the attitude root and theme of $rev$.

## 3 JITSUPEER Dataset

We describe JITSUPEER, which allows for attitude and theme-guided rebuttal generation by linking attitude roots and themes in peer reviews to canonical rebuttals based on particular rebuttal actions. To facilitate the building of JITSUPEER, we draw on existing corpora, and on established concepts, which we recast to attitude roots and attitude themes. We describe our selection of datasets (cf. §3.1) and then detail how we employ those for building JITSUPEER (cf. §3.2). We discuss licensing information in the Appendix (cf. Table 5).

### 3.1 Starting Datasets

**DISAPERE (Kennard et al., 2022).** The starting point for our work consists of reviews and corresponding rebuttals from the International Conference on Learning Representation (ICLR)[3] in 2019 and 2020. We reuse the review and rebuttal texts, which are split into individual sentences ($9,946$ review sentences and $11,103$ rebuttal sentences), as well as three annotation layers:

*Review Aspect and Polarity.* Individual review sentences are annotated as per ACL review guidelines along with their polarity (*positive, negative, neutral*) (Kang et al., 2018; Chakraborty et al., 2020). We focus on *negative* review sentences, as these are the ones rebuttals need to respond to ($2,925$ review sentences and $6,620$ rebuttal sentences). As **attitude roots**, we explore the use of the reviewing aspects – we hypothesize that reviewing aspects represent the scientific values shared by the community, e.g., papers need to be *clear*, authors need to *compare* their work, etc.

*Review–Rebuttal Links.* Individual review sentences are linked to rebuttal sentences that answer those review sentences. We use the links for retrieving candidates for canonical rebuttals.

³https://iclr.cc/Conferences/2019 https://iclr.cc/Conferences/2020

*Rebuttal Actions.* Rebuttal sentences are directly annotated with the corresponding rebuttal actions. We show the labels in the Appendix (cf. Table 7).

**PEER-REVIEW-ANALYZE (Ghosal et al., 2022).** The second dataset we employ is a benchmark resource consisting of $1,199$ reviews from ICLR 2018 with $16,976$ review sentences. Like DIS-APERE, it comes with annotation layers out of which we use a single type, *Paper Sections*. These detail to which particular part of a target paper a review sentence is referring to (e.g., *method, problem statement*, etc.). We hypothesize that these could serve as **attitude themes**. Our intuition is that while our attitude roots already represent the underlying beliefs about a work (e.g., *comparison*), the target sections add important thematic information to these values. For instance, while missing comparison within the *related work* might lead the reviewer to question the research gap, missing comparison within the *experiment* points to missing baseline comparisons. We show the paper section labels in the Appendix (cf. Table 6).

### 3.2 Enrichment

Our final goal is to obtain a corpus in which review sentences are annotated with attitude roots and themes that, in turn, link to canonical rebuttal sentences given particular rebuttal actions. As detailed above, DISAPERE already provides us with review sentences, their attitude roots (i.e., review aspects), and links to corresponding rebuttal sentences annotated with actions. The next step is to further enrich DISAPERE. For this, we follow a three-step procedure: (1) predict attitude themes using PEER-REVIEW-ANALYZE, (2) describe attitude root and theme clusters (automatically and manually), (3) identify a suitable canonical rebuttal through pairwise annotation and ranking. The enrichment pipeline is outlined in Figure 2.

#### 3.2.1 Theme Prediction

For finer-grained typing of the attitude roots, we train models on PEER-REVIEW-ANALYZE and predict themes (i.e., paper sections) on our review sentences. We test general-purpose models as well as variants specialized to the peer review domain.

**Models and Domain Specialization.** We start with BERT (Devlin et al., 2019) and RoBERTa (Liu et al., 2019), two popular general-purpose models, as well as SciBERT (Beltagy et al., 2019), which is specifically adapted for scientific text. We compare

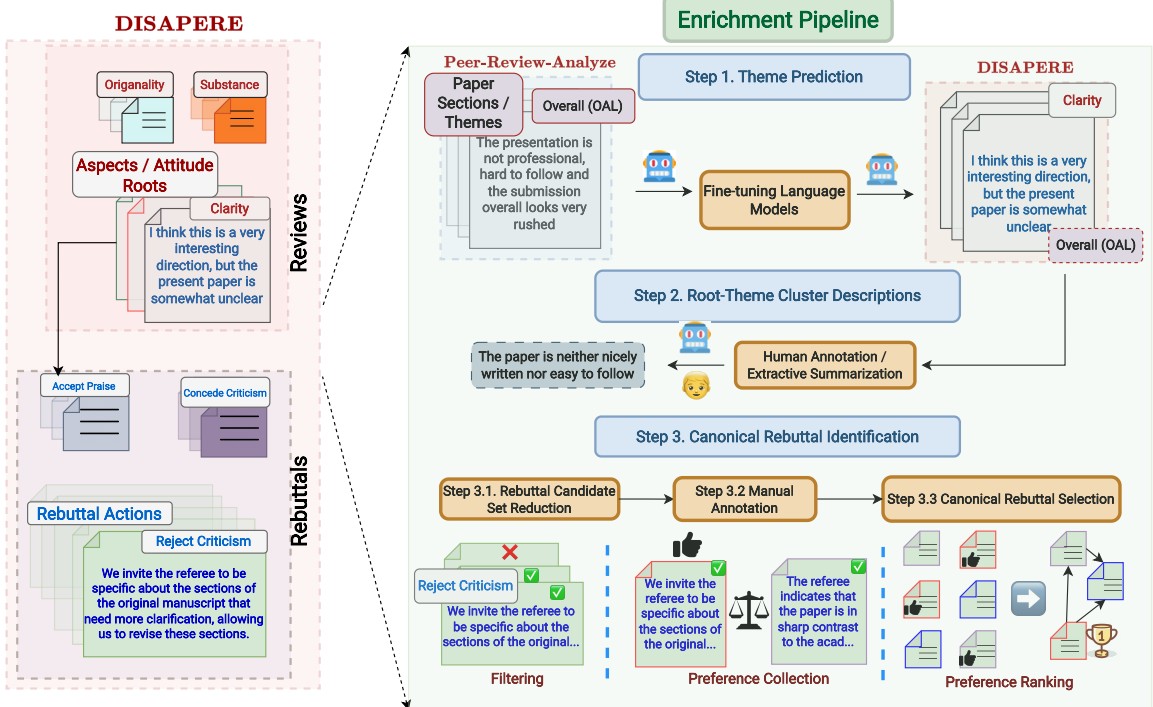

Figure 2: Structure of our starting dataset, DISAPERE along with the **Enrichment Pipeline** used for the construction of our dataset, JITSUPEER. The **Enrichment Pipeline** consists of three main steps that finally link the review sentences from each attitude root-theme cluster to the canonical rebuttals for different rebuttal actions. The three main steps are in **blue boxes** and the sub-steps are in **yellow boxes**.

those to variants, which we further specialize ourselves to the fine-grained domain of peer reviews using Masked Language Modeling (MLM) (Gururangan et al., 2020; Hung et al., 2022).

**Configuration, Training, and Optimization.** For domain specialization, we conduct intermediate MLM either on all DISAPERE reviews (suffix 'ds_all') or on the negative sentences only (suffix 'ds_neg'). We grid search over learning rates $\lambda \in \{1 \cdot 10^{-4}, 5 \cdot 10^{-5}\}$ and batch sizes $b \in \{16, 32\}$. We train for 10 epochs with early stopping (validation loss, patience of 3 epochs). For the theme prediction, a multi-label classification task, we place a sigmoid classification head on top of the transformers and fine-tune on PEER-REVIEW-ANALYZE (70/10/20 train–validation–test split) for a maximum of 10 epochs. We grid search over different learning rates ($\lambda \in \{1 \cdot 10^{-5}, 2 \cdot 10^{-5}, 3 \cdot 10^{-5}\}$) and use early stopping based on the validation performance with a patience of 3 epochs.

**Evaluation.** On PEER-REVIEW-ANALYZE, we conduct 5 runs starting from different random seeds and report the average performance across those runs. We select the model with the best micro-F1 for the final prediction. We compare against a

| Model | Precision | Recall | Micro-F1 |
|---|---|---|---|
| RANDOM | 49.48 ± .001 | 09.87 ± .001 | 16.14 ± .001 |
| MAJORITY | 33.24 ± .000 | 49.48 ± .000 | 39.76 ± .000 |
| BERT | 71.92 ± .010 | 57.05 ± .007 | 63.63 ± .003 |
| RoBERTa | 70.56 ± .013 | 59.49 ± .008 | 64.55 ± .003 |
| SciBERT | 72.64 ± .006 | 56.92 ± .005 | 64.45 ± .004 |
| BERT$_{ds\_neg}$ | 72.03 ± .107 | 58.15 ± .006 | 64.35 ± .004 |
| RoBERTa$_{ds\_neg}$ | 71.03 ± .005 | 59.52 ± .005 | 64.77 ± .002 |
| SciBERT$_{ds\_neg}$ | 72.66 ± .005 | 58.53 ± .003 | **64.83** ± .003 |
| BERT$_{ds\_all}$ | 72.38 ± .004 | 57.36 ± .007 | 64.00 ± .002 |
| RoBERTa$_{ds\_all}$ | 69.93 ± .104 | 60.47 ± .013 | 64.86 ± .005 |
| SciBERT$_{ds\_all}$ | 68.89 ± .005 | 60.61 ± .008 | 64.49 ± .004 |

Table 1: Results for general-purpose and domain-specialized models on theme enrichment task over 5 random runs. We report Precision, Recall, and Micro-F1 on the PEER-REVIEW-ANALYZE test set and highlight the best result in **bold**. We underline the 4 best performing models and separate model variants with dashes.

MAJORITY and a RANDOM baseline.

**Results and Final Prediction.** All transformer models outperform the baseline models by a huge margin (cf. Table 1). SciBERT$_{ds\_neg}$ yields the best score outperforming its non-specialized counterpart by more than 1 percentage point. This points to the effectiveness of our domain specialization. Accordingly, we run the final theme prediction for JITSUPEER with the fine-tuned SciBERT$_{ds\_neg}$. For en-

suring high quality of our data set, we only preserve predictions where the sigmoid-based confidence is higher than 70%. This way, we obtain 2,332 review sentences annotated with attitude roots and attitude themes linked to 6,614 rebuttal sentences. This corresponds to 143 clusters.

### 3.2.2 Root–Theme Cluster Descriptions

We add additional natural language descriptions to each attitude root–theme cluster. While these are not necessary for performing the final generation, they provide better human interpretability than just the label tuples. We compare summaries we obtain automatically with manual descriptions.

**Summary Generation.** For the human-written descriptions, we display $\leq 5$ sentences randomly sampled from each review cluster to a human labeler. We then ask the annotator to write a short abstractive summary (one sentence, which summarizes the shared concerns mentioned in the cluster sentences). For this task, we ask a CS Ph.D. student who has experience in NLP and is familiar with the peer review process. For the automatic summaries, we simply extract the most representative review sentence for each cluster. To this end, we embed the sentences with our domain-specific SciBERT$_{ds\_neg}$ (averaging over the last layer representations). Following Moradi and Samwald (2019), the sentence with the highest average cosine similarity with all other review sentences is the one we extract.

**Evaluation.** Having a manual and an automatic summary for each cluster in place, we next decide which one to choose as our summary. We show the summaries together with the same $\leq 5$ cluster sentences to an annotator and ask them to select the one, which better describes the cluster. We develop an annotation interface based on INCEpTION (Klie et al., 2018) and hire two additional CS Ph.D. students for the task.[4] All instances are labeled by both annotators. We measure the inter-annotator agreement on the 99 instances and obtain a Cohen's $\kappa$ of 0.8. Afterwards, a third annotator with high expertise in NLP and peer reviewing resolves the remaining disagreements.

### 3.2.3 Canonical Rebuttal Identification

Last, we identify canonical rebuttals for each attitude root-theme cluster given particular rebuttal actions. To this end, we follow a three-step procedure: first, as theoretically all 6,614 rebuttal sentences

(relating to 2,332 review sentences) are candidates for canonical rebuttals, we reduce the size of our candidate set using scores obtained from two-types of suitability classifiers. Afterwards, we manually compare pairs of rebuttal sentences from the reduced set of candidates. Finally, we compute ranks given the pair-wise scores and select the top-ranked candidate as the canonical rebuttal.

**Candidate Set Reduction.** To reduce the set of canonical rebuttal candidates to annotate in a later step, we obtain scores from two predictors: the first filter mechanism relies on confidence scores from a binary classifier, which predicts a rebuttal sentence's overall suitability for serving as a canonical rebuttal and which we train ourselves. Second, as the prototypical nature of canonical rebuttals plays an important role, we additionally use specificity scores from SPECIFICITELLER (Li and Nenkova, 2015). (1) For training our own classifier, we annotate 500 randomly selected sentences for their suitability as canonical rebuttals. The annotator is given our definition and has to decide whether or not, independent from any reference review sentences, the rebuttal sentence could potentially serve as a canonical rebuttal.[5] We then use this subset for training classifiers based on our general-purpose and domain-specific models with sigmoid heads. We evaluate the models with 5-fold cross-validation (70/10/20 split). We train all models for 5 epochs, batch size 16, and grid search for the learning rates $\lambda \in \{1 \cdot 10^{-5}, 2 \cdot 10^{-5}, 3 \cdot 10^{-5}\}$. The results are depicted in Table 3. As SciBERT$_{ds\_neg}$ achieves the highest scores, we choose this model for filtering the candidates: we predict suitabilities on the full set of rebuttals and keep only those, for which the sigmoid-based confidence score is >95%.

(2) SPECIFICITELLER, a pre-trained feature-based model, provides scores indicating whether sentences are generic or specific. We apply the model to our 500 annotated rebuttal sentences and observe that good candidates obtain scores between $0.02 - 0.78$, indicating lower specificity. We thus use this range to further reduce the number of pairs for rebuttal annotation. The complete filtering procedure leaves us with 1,845 candidates.

**Manual Annotation.** For manually deciding on canonical rebuttals given the pre-filtered set of candidates, we devise the following setup: we show a set of $\leq 5$ review sentences from an attitude root

---

[4] The annotation interface is presented in Appendix 7.4

[5] As we will only use this classifier as a rough filter, we do not doubly-annotate this data.

(e.g., *'Clarity'*) and attitude theme (e.g., *'Overall'*) cluster. We additionally pair this information with a particular rebuttal action (e.g., *'Reject Criticism'*). Next, we retrieve two random rebuttal sentences that (a) are linked to any of the review sentences in that cluster, and (b) correspond to the rebuttal action selected. The annotators need to select the best rebuttal from this pair (again, interface implemented with INCEpTION), which is a common setup for judging argument quality and ranking (e.g., Habernal and Gurevych, 2016; Toledo et al., 2019).[6] For a set of $n$ rebuttal sentences available for a particular (attitude root, attitude theme, rebuttal action)-tuple, the pairwise labeling setup requires judgments for $n(n-1)/2$ pairs (in our case, $4,402$). We recruit 2 CS Ph.D. students for this task. In an initial stage, we let them doubly annotate pairs for two attitude roots (*Clarity*, *Meaningful Comparison*). We obtain Cohen's $\kappa$ of $0.45$ (a moderate agreement, which is considered decent for such highly subjective tasks (Kennard et al., 2022)). We calculate MACE-based competencies (Hovy et al., 2013) and choose the annotator with the higher competence ($0.82$) to complete the annotations.

**Canonical Rebuttal Selection.** Following Habernal and Gurevych (2016), we obtain the best rebuttals from the collected preferences based on *Annotation Graph Ranking*. Concretely, we create a directed graph for each root–theme–action cluster with the rebuttal sentences as the nodes. The edge directions are based on the preferences: if $A$ is preferred over $B$, we create $A \rightarrow B$. We then use PageRank (Page et al., 1999) to rank the nodes (with each edge having a weight of $0.5$). The lowest-ranked nodes are the canonical rebuttals – the node has very few or no incoming edges.

### 3.3 Dataset Analysis

The final dataset consists of $2,332$ review sentences, labeled with $8$ attitude roots (aspects in DISAPERE) and $143$ themes (paper sections in PEER-REVIEW-ANALYZE). We show label distributions in Figures 3 and 4, and provide more analysis in the Appendix. Most review sentences have the attitude root *Substance*, which, also, has the highest number of themes (29). The most common theme is *Methodology* followed by *Experiments* and *Related Work*. This is intuitive since reviewers in machine learning are often concerned with the

[6]The annotation interface is presented in Appendix 7.4

| Model | Mac-F1 | Acc |
|---|---|---|
| RANDOM | $46.9 \pm 0.05$ | $56.4 \pm 0.06$ |
| MAJORITY | $71.3 \pm 0.04$ | $83.2 \pm 0.02$ |
| BERT | $93.6 \pm 0.06$ | $97.0 \pm 0.04$ |
| RoBERTa | $95.0 \pm 0.06$ | $97.0 \pm 0.02$ |
| SciBERT | $95.0 \pm 0.06$ | $96.0 \pm 0.04$ |
| BERT$_{ds\_neg}$ | $94.7 \pm 0.05$ | $96.0 \pm 0.03$ |
| RoBERTa$_{ds\_neg}$ | $93.2 \pm 0.07$ | $95.4 \pm 0.05$ |
| SciBERT$_{ds\_neg}$ | $\mathbf{96.0} \pm \mathbf{0.06}$ | $\mathbf{97.0} \pm \mathbf{0.04}$ |
| BERT$_{ds\_all}$ | $94.9 \pm 0.06$ | $95.6 \pm 0.04$ |
| RoBERTa$_{ds\_all}$ | $94.4 \pm 0.05$ | $96.2 \pm 0.04$ |
| SciBERT$_{ds\_all}$ | $96.0 \pm 0.06$ | $97.0 \pm 0.06$ |

Table 2: Performance of different models on the canonical rebuttal identification task in terms of Macro-F1 (Mac-F1) and Accuracy (Acc) averaged over 5 folds.

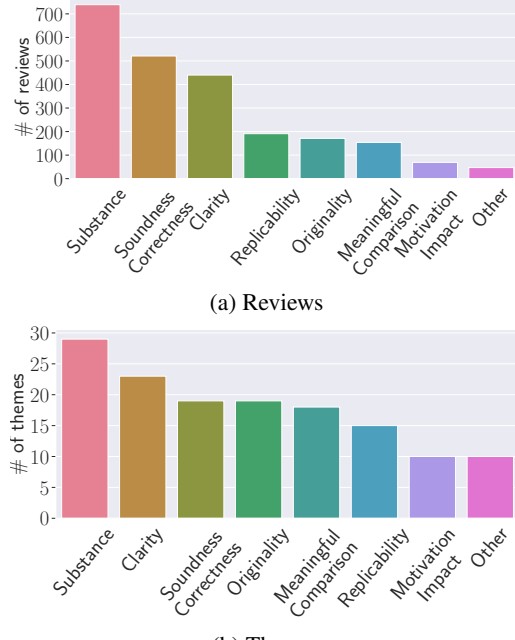

(a) Reviews

(b) Themes

Figure 3: Final data set analysis: number of review sentences and number of themes per attitude root.

soundness and utility of the methodology. In total, we identified 302 canonical rebuttals for different attitude roots and rebuttal actions.[7] Our canonical rebuttals can be mapped to $2,219$ review sentences (out of $2,332$). The highest number of canonical rebuttal sentences relate to the rebuttal action *Task Done* and the attitude root *Substance*. In Table 3, we show examples of some of the canonical rebuttals. We clearly note that different attitude root–theme descriptions connect to different canonical rebuttals (e.g., *Concede Criticism* in *Clarity* and *Substance*).

[7]Out of these, we obtained 117 without annotation since these were the only rebuttal sentences left after reducing the candidate set.

| Attitude Roots | Description | Canonical Rebuttals | |
|---|---|---|---|
| Clarity | The paper is neither nicely written nor easy to follow. | *Answer*: 'We have significantly improved the writing, re-done the bibliography and organized the most important theorems into a clearer presentation.' | *Concede Criticism*: 'In addition, there are indeed a few places in the paper where our phrasing could have been better, thank you for pointing this out.' |
| | Unclear Description of Method | *Future Work*: 'While we have provided some diagnostics statistics, understanding this method deeply that will help fuel interesting future research.' | *Reject Criticism*: 'As far as explaining the method of combination, and the associated mathematical properties, we have tried to do this in greater detail in section 3 (Approach).' |
| Substance | Incomplete details on performance of the method | *Structuring*: 'The experiment section lacks more detailed analysis which can intuitively explain how well the proposed method performs on the benchmarks' | *Answer*: 'While we put this experiment into the appendix , for now, to not change the main paper too much compared to the submitted version, if the reviewers agree we would also be very happy to include this experiment in the main paper.' |
| | Limited improvement over baselines | *Task Done*: 'We provided a detailed explanation about the experimental setting and further experimental results of the state-of-the-art performance in our response to "The Common concerns about experimental setting and results".' | *Concede Criticism*: 'Nevertheless, we agree with your comments that it is more meaningful to emphasize our improvement over the state-of-the-art training methods.' |

Table 3: Canonical rebuttals for different rebuttal actions (in *italics*), attitude roots, and theme descriptions.

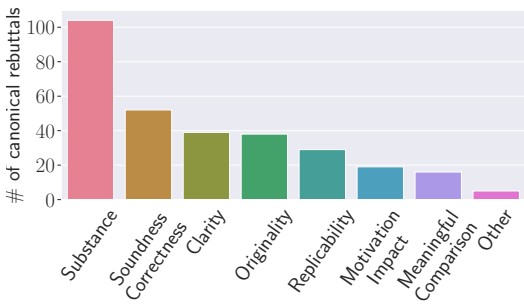

Figure 4: Final dataset analysis: number of canonical rebuttals per attitude root.

| Models | r | MAP | MRR |
|---|---|---|---|
| RANDOM | 0.010 | $13.86 \pm 0.01$ | $13.90 \pm 0.01$ |
| BERT | 0.241 | $33.56 \pm 0.01$ | $35.59 \pm 0.01$ |
| RoBERTa | 0.447 | $35.41 \pm 0.01$ | $35.40 \pm 0.01$ |
| SciBERT | 0.470 | $36.73 \pm 0.01$ | $36.72 \pm 0.01$ |
| BERT$_{ds\_neg}$ | 0.383* | $\mathbf{39.20} \pm 0.02$ | $\mathbf{39.13} \pm 0.02$ |
| RoBERTa$_{ds\_neg}$ | 0.454 | $36.21 \pm 0.01$ | $36.21 \pm 0.01$ |
| SciBERT$_{ds\_neg}$ | 0.468 | $36.67 \pm 0.01$ | $36.68 \pm 0.01$ |
| BERT$_{ds\_all}$ | 0.488* | $38.37 \pm 0.03$ | $38.37 \pm 0.03$ |
| RoBERTa$_{ds\_all}$ | 0.323* | $37.82 \pm 0.03$ | $37.67 \pm 0.03$ |
| SciBERT$_{ds\_all}$ | **0.491*** | $35.98 \pm 0.01$ | $36.00 \pm 0.01$ |

Table 4: Results of non-specialized and specialized variants of BERT, RoBERTa, and SciBERT on the canonical rebuttal scoring Task. We report Pearson's Correlation Coefficient (**r**), Mean Average Precision (MAP) and Mean Reciprocal Rank (MRR) averaged over 5 runs. (*) statistically significant differences (p < 0.05).

# 4 Baseline Experiments

Along with end-to-end canonical rebuttal generation, we propose three novel tasks on our dataset.

## 4.1 Canonical Rebuttal Scoring

**Task Definition.** Given a natural language description $d$, and a rebuttal action $a$, the task is to predict, for all rebuttals $r \in R$ (relating to particular attitude root–theme cluster), a score indicating the suitability of $r$ for serving as the canonical rebuttal for that cluster.

**Experimental Setup.** The task amounts to a *regression problem*. We only consider combinations of rebuttal actions and attitude root–theme clusters that have a canonical rebuttal (50 attitude root–theme cluster descriptions with $3,986$ rebuttal sentences out of which 302 are canonical). We use the PageRank scores from before (cf. §3.2.3) as our prediction target for model training.[8] To avoid any information leakage, we split the data into

train-validation-test on the level of the number of attitude roots $(4 - 1 - 3)$. The total number of instances amounts to $5941$ out of which we use $2723 - 1450 - 1768$ for the train-validation-test. We experiment with all models described in §3.2.1 in a fine-tuning setup. Following established work on argument scoring (e.g., Gretz et al., 2020; Holtermann et al., 2022), we concatenate the description $d$ with the action $a$ using a separator token (e.g., [SEP] for BERT). We grid search for the optimal number of epochs $e \in \{1, 2, 3, 4, 5\}$ and learning rates $\lambda \in \{1 \cdot 10^{-4}, 2 \cdot 10^{-4}\}$ with a batch size $b = 32$ and select models based on their performance on the validation set. We also compare to a baseline, where we randomly select a score between $0$ and $1$. We report Pearson's correlation coefficient (**r**), and, since the scores can be used for ranking, Mean Average Precision (MAP) and Mean Reciprocal Rank (MRR).

**Results.** From Table 4, we observe that most of the domain-specialized models perform better than their non-specialized counterparts. SciBERT$_{ds\_all}$

---

[8]For canonical rebuttals obtained without pairwise annotation (these rebuttals were the only ones being predicted as canonical by our canonical rebuttal identifier and had no other candidates for comparison) we set the score to 0 (since the lower the score, the better the rebuttal)). Vice versa, for all other rebuttals, the score is set to 1.

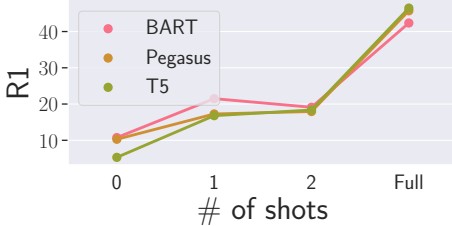

Figure 5: ROUGE-1 variation on the Review Description Generation task of BART, Pegasus, and T5 with an increasing number of shots.

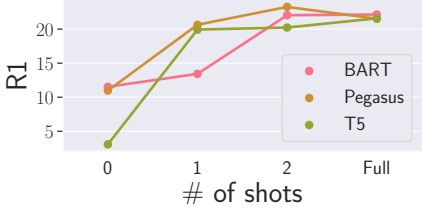

Figure 6: ROUGE-1 variation on the End2End Reuttal Generation task of BART, Pegasus, and T5 with an increasing number of shots.

has the highest Pearson correlation across the board, however, $BERT_{ds\_neg}$ has the highest ranking scores. The use of other cluster-related information such as representative review sentences, and paraphrasing the descriptions may lead to further gains which we leave for future investigation.

## 4.2 Review Description Generation

**Task Definition.** Given a peer review sentence $rev$, the task is to generate the abstract description $d$ of the cluster to which $rev$ belongs to.

**Experimental Setup.** Our data set consists of $2,332$ review sentences each belonging to one of $144$ clusters with associated descriptions. We apply a train–validation–test split $(70/10/20)$ and experiment with the following seq2seq models: BART (Lewis et al., 2020) (bart-large), Pegasus (Zhang et al., 2020) (pegasus-large), and T5 (Raffel et al., 2020) (t5-large). We grid search over the number of epochs $e \in \{1, 2, 3, 4, 5\}$ learning rates $\lambda \in \{1 \cdot 10^{-4}, 5 \cdot 10^{-4}, 1 \cdot 10^{-5}\}$ with a batch size $b = 32$. We use beam search with 5 beams as the decoding strategy. We run this experiment in a full fine-tuning setup as well as in zero- and few-shot scenarios (with random shot selection). We report the performance in terms of lexical overlap and semantic similarity (ROUGE-1 (R-1), ROUGE-2 (R-2), ROUGE-L (R-L) (Lin, 2004), and BERTscore (Zhang* et al., 2020)).[9]

**Results.** We show the R-1 scores in Figure 5 (full results in Table 11). Interestingly, all models exhibit a very steep learning curve, roughly doubling their performance according to most measures when seeing a single example only. BART excels in the 0-shot and 1-shot setup, across all scores. However, when fully fine-tuning the models, T5 performs best. We hypothesize that this relates to T5's larger capacity (406M params in BART vs. 770M params in T5).

---

[9] We use the default *roberta-large* model for evaluation.

## 4.3 End2End Canonical Rebuttal Generation

**Task Definition.** Given a review sentence $rev$, and a rebuttal action $a$, the task is to generate the canonical rebuttal $c$.

**Experimental Setup.** We start from $2,219$ review sentences, which have canonical rebuttals for at least 1 action. As input, we concatenate $rev$ with $a$ placing a separator token in between resulting in $17,873$ unique review–rebuttal action instances. We use the same set of hyperparameters, models, and measures as before (cf. §4.2) and experiment with full fine-tuning, and zero-shot as well as few-shot prediction. For these experiments, we apply a $70/10/20$ splits for obtaining train–validation–test portions on the level of the canonical rebuttals ($302$ rebuttals linked to $17,873$ unique instances).

**Results.** The differences among the models are in-line with our findings from before (Figure 6, full results in Table 12): BART excels in the zero-shot and few-shot setups, and T5 starts from the lowest performance but quickly catches up with the other models. However, the models' performances grow even steeper than before, and seem to reach a plateau already after two shots. We think that this relates to the limited variety of canonical rebuttals and to the train–test split on the canonical rebuttal level we decided to make – the task is to generate templates, and generalize over those. With seeing only few of those templates, the models quickly get the general gist, but are unable to generalize beyond what they have been shown. This finding leaves room for future research and points at the potential of data efficient approaches in this area.

## 5 Related Work

**Peer Review Analysis.** In recent years, research on peer reviews has gained attention in NLP, with most of the efforts concentrated on creating new datasets to facilitate future research. For instance, Hua et al. (2019) presented a data set for analyzing discourse

structures in peer reviews annotated with different review actions (e.g., EVALUATE, REQUEST). Similarly, Fromm et al. (2020) developed a dataset that models the stance of peer-review sentences in relation to accept/reject decisions. Yuan et al. (2022) extended peer review labels with polarity labels and aspects based on the ACL review guidelines (as in (Chakraborty et al., 2020)). Newer works focused mainly on linking the review sections to the exact parts of the target papers they relate to (Ghosal et al., 2022; Kuznetsov et al., 2022; Dycke et al., 2023). Overall, only few works focused on understanding the dynamics between review and rebuttal sentences. Exceptions to this are provided by Cheng et al. (2020) and Bao et al. (2021) who study discourse based on sentence-wise links between review and rebuttal sentences. Kennard et al. (2022) proposed a dataset that unifies existing review annotation labels and also studied review-rebuttal interaction.

**Computational Argumentation and Attitude Roots.** Computational argumentation covers the mining, assessment, generation and reasoning over natural language argumentation (Lauscher et al., 2022b). Here, most works focused on analyzing the explicated contents of an argument, e.g., w.r.t. its quality (Toledo et al., 2019; Gretz et al., 2020), or structure (Morio et al., 2020), and on generating arguments based on such surface-level reasoning (Slonim et al., 2021). In contrast, Hornsey and Fielding (2017) analyze the underlying reasons driving peoples arguments. In a similar vein, Lewandowsky et al. (2013) study the motivated rejection of science and demonstrate similar attitudes toward climate-change skepticism. Fasce et al. (2023) extend this theory to the domain of anti-vaccination attitudes during the COVID-19 era. We borrow this idea and adapt it to the domain of peer reviewing to understand scientific attitudes.

## 6    Conclusion

In this work, we explored Jiu-Jitsu argumentation for peer reviews, based on the idea that reviewers' comments are driven by their underlying attitudes. For enabling research in this area, we created JITSUPEER, a novel data set consisting of review sentences linked to canonical rebuttals, which can serve as templates for writing effective peer review rebuttals. We proposed different NLP tasks on this dataset and benchmarked multiple baseline strategies. We make the annotations for JITSUPEER

publicly available. We believe that this dataset will serve as a valuable resource to foster further research in computational argumentation for writing effective peer review rebuttals.

## Limitations

In this work, we present a novel resource in the domain of peer reviews, JITSUPEER. Even though we develop our data set with the goal of fostering equality in science through helping junior researchers and non-native speakers with writing rebuttals in mind, naturally, this resource comes with a number of limitations: JITSUPEER contains different attitude roots and attitude themes along with the canonical rebuttals derived from the peer reviews from ICLR 2019 and 2020. ICLR is a top-tier Machine Learning Conference and thus the taxonomy developed in this process is specific to Machine Learning Community and **does not cover** the peer reviewing domain completely (e.g., natural sciences, arts, and humanities, etc). Thus, the resource will have to be adapted as the domain varies. The canonical rebuttals also do not form a closed set since the original dataset DISAPERE, from which we started, does not contain rebuttals for every review sentence. Accordingly, the peer review to canonical rebuttal mapping is sparse. We therefore, and for other reasons, highlight that writing rebuttals should be a human-in-the-loop process where models trained on JITSUPEER can provide assistance by generating templates that can be further refined for writing customized rebuttals.

## Acknowledgements

This work has been funded by the German Research Foundation (DFG) as part of the Research Training Group KRITIS No. GRK 2222. The work of Anne Lauscher is funded under the Excellence Strategy of the German Federal Government and the Federal States.

We thank Aniket Pramanick, Nils Dycke, Luke Bates, and Haishuo Fang for their valuable feedback and suggestions on a draft of this paper. We would also like to extend our gratitude to Jing Yang, Tilman Beck, Cecilia Liu, Maral Dadvar, and Inderdeep Minhas for their help and support in annotating some portions of our dataset.

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

| Dataset | License |
|---|---|
| DISAPERE | Creative Commons Attribution-NonCommercial 4.0 International Public License |
| PEER-REVIEW-ANALYZE | MIT License |
| JITSUPEER | Creative Commons Attribution-NonCommercial 4.0 International Public License |

Table 5: License details of the different datasets used in the paper.

# 7 Appendix

## 7.1 Information on dataset licenses

The licenses for different datasets used in the paper are listed in Table 5.

## 7.2 Dataset details

We list the distribution of review sentences (in %) for the datasets used in our work namely, DIS-APERE and PEER-REVIEW-ANALYZE in Table 6. We show the distribution of rebuttal sentences with respect to rebuttal actions in Table 7. We display part of the label hierarchy developed for JITSU-PEER in Table 8. We plot the number of canonical rebuttals with respect to different rebuttal actions in Fig 9. *'Task Done'* and *'Answer'* are the most common rebuttal actions for the annotated canonical rebuttals.

## 7.3 Probing of the Domain Specialized Models

In order to gauge the quality of review representations of the different transformer models (pre-trained and domain specialized), we additionally perform a probing task. We first start with describing the **probing** task. We adopt a classic probing procedure following previous works (Tenney et al., 2019; Zhang et al., 2021; Lauscher et al., 2022a) where a classifier is placed on top of the frozen features. Following (Zhang et al., 2021; Lauscher et al., 2022a), we use a simple two-layer feed-forward neural network on top of the frozen features. For a given text, we average over the representation of all the tokens except the special tokens. We perform the probing task on DISAPERE. We use the already available classification tasks in the dataset such as 'Aspect', 'Review Action', and 'Fine-grained Review Action' classification as the proxy tasks. We evaluate the models in terms of Accuracy (Acc) and Macro-F1 (F1) metrics. For

| Datasets | Review Labels | % of rev sents |
|---|---|---|
| DISAPERE | Soundness-Correctness | 22.15 |
| | Clarity | 21.26 |
| | Substance | 30.52 |
| | Originality | 7.24 |
| | Replicability | 7.48 |
| | Motivation- Impact | 3.15 |
| | Meaningful-Comparison | 5.81 |
| | Other | 2.39 |
| PEER-REVIEW-ANALYZE | Methodology | 27.73 |
| | Introduction | 2.84 |
| | Related Work | 5.05 |
| | Problem Definition | 3.43 |
| | Datasets | 1.49 |
| | Experiments | 4.72 |
| | Results | 3.51 |
| | Tables & Figures | 2.22 |
| | Analysis | 0.90 |
| | Future Work | 0.43 |
| | Overall | 6.82 |
| | Bibliography | 0.99 |
| | External | 1.41 |

Table 6: Distribution of review sentences in DIS-APERE and PEER-REVIEW-ANALYZE. For PEER-REVIEW-ANALYZE, these labels constitute 60.24% of review sentences, the rest of the review sentences are annotated with different combinations of these labels.

**probing**, we follow (Lauscher et al., 2022a), and use a batch size of 32 with a learning rate, $\lambda = 1e-3$ using the Adam optimizer. The models are trained for 100 epochs with early stopping on the validation set with patience of 5. The results are averaged over 5 runs. We use the standard train/dev/test splits from DISAPERE.

From Table 9, we observe that the best-performing models are pre-trained RoBERTa, SciBERT, SciBERT$_{ds\_neg}$, and SciBERT$_{ds\_all}$.

## 7.4 Annotator Details and Interface Design

We recruited 2 CS Ph.D. students for the different tasks that require human annotation while creating our dataset, JITSUPEER. Initially, we experimented with varying degrees of expertise but found that CS Masters students struggled with the task, which is why we resorted to recruiting PhD students. We trained the annotators initially for 1 hour to explain the guidelines and then answered their questions when needed. We adapted the INCEpTION platform (Klie et al., 2018) for carrying out annotations. In the dataset creation pipeline, there are two tasks that explicitly require human intervention: i) Root-Theme Cluster Description (cf. §3.2.2) and, ii) Manual Evaluation for Canonical Rebuttal Identification (cf. §3.2.3). The interfaces for both of these tasks are presented in Fig 7 and Fig 8 respec-

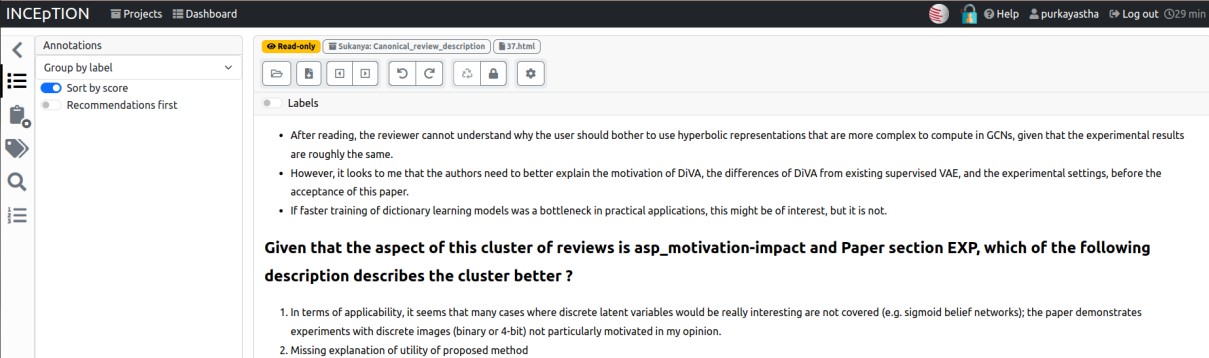

Figure 7: Snapshot of the interface used for collecting annotations for the Root-Theme Cluster Description Task. The attitude root here is *'Motivation Impact'* (asp_motivation-impact) and the theme *'Experiment'*(EXP).

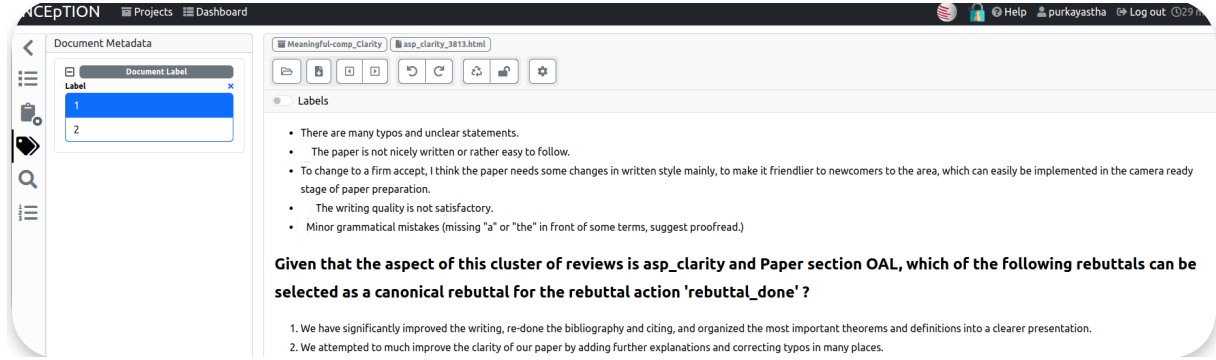

Figure 8: Snapshot of the interface used for the preference collection for the Manual Evaluation for Canonical Rebuttal Identification Task. The attitude root *'Clarity'* (asp_clarity) and the theme *'Overall'*(OAL) with respect to the rebuttal action, *'rebuttal done'*.

| Rebuttal Actions | % of rebuttal sents |
|---|---|
| Answer | 33.93 |
| Reject criticism | 15.57 |
| Structuring | 15.79 |
| The task is done | 9.70 |
| Concede criticism | 3.69 |
| The task will be done | 2.11 |
| Accept for future work | 1.28 |
| Accept praise | 0.05 |
| Mitigate criticism | 3.31 |
| Reject request | 1.13 |
| Refute question | 1.27 |
| Contradict assertion | 0.76 |
| Summary | 9.86 |
| Social | 1.07 |
| Followup questions | 0.32 |
| Other | 0.16 |

Table 7: Distribution of rebuttal sentences (in %) for different rebuttal actions in DISAPERE

tively.

## 7.5 Canonical Rebuttal Ranking Task

**Task Definition.** As another variant of the Canonical Rebuttal Scoring Task (cf §4.1), we now seek to directly identify the canonical rebuttal using retrieval: given a description $d$ and a rebuttal action $a$, the task is to retrieve the canonical rebuttal $c$ from the set of rebuttals $R$ that corresponds to the attitude root—theme cluster.

**Experimental Setup.** We formulate this problem as a *ranking task*. The number of cluster descriptions and rebuttals is the same as in the Canonical Rebuttal Scoring Task (cf §4.1). We implement BM25 based information-retrieval system as a baseline for this task. We also use different bi-encoder-based models built using S-BERT (Reimers and Gurevych, 2019). As in Task 1, we split the data on the level of attitude roots $(4 - 1 - 3)$ for training–validation–test. We concatenate $d$ and $a$ using [SEP] token. We encode the concatenated cluster description and rebuttals independently and use cosine similarity as a distance measure for training and inference. We finetune pre-trained sentence transformer models on this task along with training new sentence transformer models using embeddings from different language models (BERT, RoBERTa, SciBERT, and their domain-specialized variants). We perform grid search over learning rates $\lambda \in \{1e{-}5, 2e{-}5\}$, epochs $e \in \{1, 2, 3, 4, 5\}$ and batch sizes, $b \in \{16, 32\}$. We report Mean Average Precision (MAP) and Mean Reciprocal Rank (MRR).

| Attitude roots | Themes | Descriptions | Example review sents. |
|---|---|---|---|
| Clarity | Overall | The paper is not nicely written or rather easy to follow | 'The paper can benefit from proofreading.', 'I think this is a very interesting direction, but the present paper is somewhat unclear' |
| | Method | Unclear description of method | 'What is dt in Algorithm 1 description ?', 'The method is very confusingly presented and requires both knowledge of HAT as well as more than one reading to understand.' |
| Originality | Experiments | Not enough novelty in experiments (seems similar to previous work) | 'Simply because for continuous variables similar experiments have been reported before', 'Also, I find the experiments done in section 3 and 4 are similar to previous works and even the conclusions are similar. ' |
| | Results | Not enough originality in results | 'As the authors admit, the main result is not especially surprising.', 'although there must be some small innovations, I thought that all the results had more or less been proven by Dupuis and co-authors:' |
| Meaningful Comparison | Related Work | Missing Baselines | 'Attacking CRBMs is highly relevant and should be included as a baseline.', 'The paper should also talk about the details of ARNet and discuss the difference, as I assume they are the most related work' |
| | Results | More comparisons needed with variations of the proposed method | 'I don't think this is really a fair comparison; I would have liked to have seen results for the unmodified reward function.' , 'The human score of 91.4% is based on majority vote, which should be compared with an ensemble of deep learning prediction.' |
| Soundness Correctness | Dataset | Claim on the datasets is questionable | ' in section 4.3, there is no guarantee that the intersection between the training set and test set is empty.', 'The test set should not used before the best compression scheme is selected.' |
| | Tables and Figures | Incomplete ablation of tables and figures | 'The uncertainties produced by CDN in Figure 2 seems strange.', 'It did also not match any numbers in Tab. 4 of the appendix.' |
| Substance | Analysis | Lack of analysis | 'Therefore, it may be a good idea for the authors to analyze the correlation between FSM changes and accuracy changes.', 'no qualitative analysis on how modulation is actually used by the systems.' |
| | Dataset | Less number of datasets used | 'BSD 500 only contains 500 images, and it would be good if more diverse set of images are considered.', 'Third, the datasets used in this paper are rather limited.' |
| Motivation Impact | Methodology | Limited insight based on design choices | 'The main drawback of the paper is that it seems to be more engineering-focused, and doesn't provide much insight into semi-supervised learning.', 'Thus we may only apply the proposed model on a few tasks with exactly known F.' |
| | Results | Limited impact of results | 'The results are overall not very impressive.', 'The results of the paper do not give major insights into what are the preferred techniques for training GANs, and certainly not why and under what circumstances they'll work.' |
| Replicability | Related Work | Missing implementation details of related work used as baselines | 'The graph neural networks used in the model are not described in the paper, only a reference to Paliwal et al (2019) is given.', 'It would be helpful to have a brief paragraph describing this architecture, for readers not familiar with the referenced paper.' |
| | Results | Missing details for reproducibility of results | 'Are the authors willing to release the code? Overall the model looks complicated and the appendix is not sufficient to reproduce the results in the paper.', 'I trust that the authors did in fact achieve these results but I cannot figure out how or why.' |
| Other | Overall | Reasons for rejection (not fit for conference, contains several weaknesses, etc.) | 'I think the paper does not fit this conference. It is better to be presented in a Demonstration section at a *ACL conference.', 'Overall, I feel that this paper falls short of what it promises, so I cannot recommend acceptance at this time.' |
| | Bibliography | Missing citations | 'The above papers are not cited in this paper.', 'The cited paper 'Learning an adaptive learning rate schedule' does not appear online.' |

Table 8: Attitude roots along with themes, descriptions, and example review sentences in our proposed dataset, JITSUPEER. The attitude roots are the same as the aspects in DISAPERE (Kennard et al., 2022) and the themes related to the paper sections proposed in PEER-REVIEW-ANALYZE (Ghosal et al., 2022).

| Model | Review Action | | Fine Review Action | | Aspect Identification | | Avg | |
|---|---|---|---|---|---|---|---|---|
| | Acc | F1 | Acc | F1 | Acc | F1 | Acc | F1 |
| BERT | 78.1 ± 0.008 | 73.9 ± 0.189 | 47.4 ± 0.012 | 37.0 ± 0.010 | 37.2 ± 0.030 | 27.4 ± 0.020 | 54.2 | 46.1 |
| RoBERTa | 78.7 ± 0.005 | 75.8 ± 0.009 | 45.4 ± 0.002 | 36.3 ± 0.009 | 42.2 ± 0.010 | 31.0 ± 0.001 | **55.4** | **47.7** |
| SciBERT | 80.2 ± 0.004 | 76.7 ± 0.005 | 45.2 ± 0.020 | 35.3 ± 0.013 | 41.5 ± 0.010 | 29.5 ± 0.020 | **55.7** | **47.2** |
| BERT$_{ds\_neg}$ | 76.8 ± 0.021 | 69.3 ± 0.057 | 43.7 ± 0.020 | 33.6 ± 0.040 | 31.8 ± 0.030 | 23.8 ± 0.020 | 50.8 | 42.3 |
| RoBERTa$_{ds\_neg}$ | 78.6 ± 0.210 | 73.3 ± 0.051 | 45.4 ± 0.020 | 35.0 ± 0.030 | 39.6 ± 0.040 | 30.6 ± 0.004 | 54.5 | 46.3 |
| SciBERT$_{ds\_neg}$ | 77.8 ± 0.034 | 74.9 ± 0.022 | 48.6 ± 0.020 | 39.5 ± 0.020 | 40.0 ± 0.020 | 30.4 ± 0.010 | **55.5** | **48.3** |
| BERT$_{ds\_all}$ | 78.2 ± 0.015 | 72.7 ± 0.040 | 48.9 ± 0.010 | 38.6 ± 0.010 | 32.9 ± 0.040 | 24.3 ± 0.020 | 53.3 | 45.2 |
| RoBERTa$_{ds\_all}$ | 78.2 ± 0.015 | 72.7 ± 0.040 | 47.3 ± 0.030 | 36.2 ± 0.040 | 40.7 ± 0.030 | 30.0 ± 0.010 | 55.4 | 46.3 |
| SciBERT$_{ds\_all}$ | 79.7 ± 0.050 | 76.4 ± 0.005 | 45.8 ± 0.050 | 36.0 ± 0.050 | 41.7 ± 0.010 | 31.8 ± 0.010 | **55.7** | **48.3** |

Table 9: Results for our experiments on the first phase of the theme prediction task - Probing. We demonstrate results for three tasks - Review Action, Fine Review Action, and Aspect Identification in terms of Accuracy (Acc) and F1. **Bold** indicates the 4 best-performing models across all the tasks on average.

| Models | MRR | MAP |
|---|---|---|
| BM25 | 0.209 ±0.011 | 0.211 ± 0.015 |
| allmini-lm[*] | 0.218 ± 0.016 | 0.219 ± 0.002 |
| paraphrase-albert[*] | 0.195 ±0.001 | 0.196 ± 0.001 |
| BERT | 0.227 ± 0.003 | 0.229 ± 0.002 |
| RoBERTa | 0.226 ± 0.001 | 0.227 ± 0.001 |
| SciBERT | 0.209 ± 0.001 | 0.209 ± 0.002 |
| BERT$_{ds\_neg}$ | **0.231** ± 0.003 | **0.232** ± 0.002 |
| RoBERTa$_{ds\_neg}$ | 0.220 ± 0.002 | 0.221 ± 0.002 |
| SciBERT$_{ds\_neg}$ | 0.220 ± 0.002 | 0.222 ± 0.002 |
| BERT$_{ds\_all}$ | 0.229 ± 0.017 | 0.230 ± 0.012 |
| RoBERTa$_{ds\_all}$ | 0.193 ± 0.018 | 0.195 ± 0.012 |
| SciBERT$_{ds\_all}$ | 0.193 ± 0.017 | 0.195 ± 0.012 |

Table 10: Results on Canonical Rebuttal Ranking Task. We report Mean Average Precision (MAP), Mean Reciprocal Rank (MRR) for this task. *ds_neg*, *ds_all* represent models that are domain specialized on all the reviews and negative reviews from DISAPERE respectively. (**\***) represents pre-trained sentence transformer models obtained from [https://www.sbert.net/index.html] and finetuned for this task.

**Results.** Table 10 shows the performance for different models. Interestingly, BM25 has a competitive performance with different finetuned models which is in line with Kennard et al. (2022). However, the best model is the finetuned sentence transformer model for BERT trained on negative reviews (BERT$_{ds\_neg}$). This is in line with our findings for Canonical Rebuttal Scoring Task (see Table 4) where BERT$_{ds\_neg}$ had the best ranking scores.

## 7.6 Review Description Generation Task

The full results for this task are provided in Table 11.

## 7.7 End2End Canonical Rebuttal Generation Task

The full results for this task are provided in Table 12.

| # shots | Models | R-1 | R-2 | R-L | BERTsc |
|---|---|---|---|---|---|
| 0 | BART | **10.75** | **2.43** | **9.31** | **0.16** |
| | Pegasus | 10.28 | 2.30 | 8.96 | **0.16** |
| | T5 | 5.29 | 0.93 | 4.87 | -0.05 |
| 1 | BART | **21.49** | 5.70 | **19.87** | **0.359** |
| | Pegasus | 17.26 | 6.06 | 16.58 | 0.314 |
| | T5 | 16.79 | 4.21 | 15.94 | 0.260 |
| 2 | BART | **19.11** | 5.26 | **18.42** | **0.342** |
| | Pegasus | 17.94 | 6.55 | 17.33 | 0.322 |
| | T5 | 18.36 | 6.14 | 17.50 | 0.276 |
| Full | BART | 42.36 | 33.64 | 42.17 | 0.496 |
| | Pegasus | 45.73 | 37.83 | **46.36** | 0.516 |
| | T5 | **46.47** | **37.85** | 45.95 | **0.520** |

Table 11: Performance of BART, Pegasus, and T5 on the Review Description Generation task. We report the zero-shot, few-shot, and full fine-tuning performance in terms of ROUGE-1 (R-1), ROUGE-2 (R-2), ROUGE-L (R-L) and BERTscore.

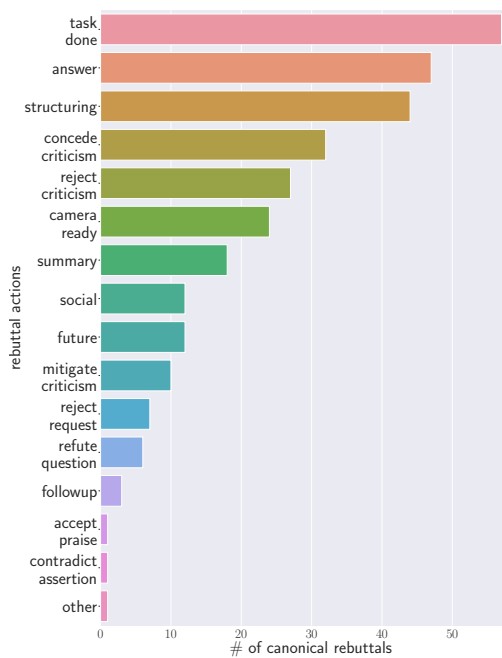

Figure 9: Final dataset analysis: number of canonical rebuttals per rebuttal action.

| # shots | Models | R-1 | R-2 | R-L | BERTscore |
|---------|--------|-----|-----|-----|-----------|
| | BART | **11.53** | **0.87** | **9.20** | **0.005** |
| 0 | Pegasus | 10.97 | 0.16 | 2.73 | -0.32 |
| | T5 | 3.08 | 0.79 | 8.88 | -0.002 |
| | BART | **13.43** | **2.40** | **11.01** | **0.132** |
| 1 | Pegasus | 20.64 | 4.36 | 16.48 | 0.249 |
| | T5 | 19.93 | 3.63 | 15.09 | 0.203 |
| | BART | **22.04** | 3.42 | **16.90** | 0.256 |
| 2 | Pegasus | 23.26 | 4.94 | 18.18 | **0.270** |
| | T5 | 20.23 | **3.59** | 16.27 | 0.224 |
| | BART | **22.14** | **3.88** | 15.67 | **0.218** |
| Full | Pegasus | 21.52 | 3.87 | 15.69 | 0.193 |
| | T5 | 21.58 | 3.86 | **15.98** | 0.195 |

Table 12: Performance of BART, Pegasus, and T5 on the End2End Canonical Rebuttal Generation task. We report the zero-shot, few-shot, and full-finetuning performance in terms of ROUGE-1 (R-1), ROUGE-2 (R-2), ROUGE-L (R-L) and BERTscore.