# OpenReview forum: "Exploring Jiu-Jitsu Argumentation for Writing Peer Review Rebuttals"
_EMNLP/2023/Conference — EMNLP 2023 Main_

### Official Review · Reviewer_b1L5 · 2023-07-24

**Soundness:** 3
**Typos Grammar Style And Presentation Improvements:** 1. In line 367, "Table 3" should be m…

**Excitement:**

4: Strong: This paper deepens the understanding of some phenomenon or lowers the barriers to an existing research direction.

**Paper Topic And Main Contributions:**

This paper explores the problem of writing effective rebuttals in peer reviews. The authors propose the use of Jiu-Jitsu argumentation, inspired by the Jiu-Jitsu combat system, as a strategy for generating rebuttals that align with an opponent's underlying beliefs. The main contributions of this paper include:

1) the introduction of attitude roots and attitude themes in argumentation;
2) the enrichment of an existing dataset for discourse structure in peer reviews with attitude roots, attitude themes, and canonical rebuttals;
3) the exploration of the potential of leveraging attitude roots and themes in peer review rebuttals.

**Questions For The Authors:**

Question A: In Section 3.2.1 Theme Prediction and Section 3.2.3 Canonical Rebuttal Identification, why don't you choose a more powerful model such as T5 or GPT for classification? Since the data consturction depends on the performance of classification, I wonder if a more powerful would be benificial.

**Reasons To Accept:**

1. This work is the first one to explore Jiu-Jitsu argumentation for peer review by proposing the novel task of attitude and theme-guided rebuttal generation.

2. This paper construction a new dataset to for the task of end-to-end attitude and theme guided rebuttal generation and two subtasks.

3. The paper also provides some benchmarks for the attitude and theme-guided rebuttal generation tasks.

**Reasons To Reject:**

1. The number of annotators used for human annotation are quite small (one or two as described in the paper), which may results in the human annotation less convincing.

2. The dataset construction depends on the model performance of classification at middle steps, which may result in potential biases.

3. Some details of the human annotation are missing. For example, the authors mention that "We develop an annotation interface based on INCEpTION (Klie et al., 2018)", but the questions for the annotation are not given.

**Reproducibility:**

4: Could mostly reproduce the results, but there may be some variation because of sample variance or minor variations in their interpretation of the protocol or method.

**Reviewer Confidence:**

3: Pretty sure, but there's a chance I missed something. Although I have a good feel for this area in general, I did not carefully check the paper's details, e.g., the math, experimental design, or novelty.

---

> ### Author Rebuttal · Authors · 2023-08-28
>
> We would like to thank the reviewer for their detailed feedback on our work. We justify the concerns below:
>
> # Reasons
> #### $R1$. **[Annotators] ”The number of annotators used for human annotation are quite small (one or two as described in the paper), which may results in the human annotation less convincing”**
> $Ans$. We thank the reviewer for sharing this important concern. However, in our case, we do not believe that having two annotators will result in lower data set quality. On the contrary: since we deal with a task in the domain of peer review, our annotations require scientific expert knowledge leading to a smaller pool of qualified annotators. Our annotator selection and our final agreement is comparable to other studies in this domain, pointing to the reliability of our dataset. For instance, Kuznetsov et. al (2022) choose two annotators and report an inter-annotator agreement (IAA) of $0.53$ (comparable to $0.45$ in our case), and Kennard et. al (2022) report an IAA range of $0.447$ - $0.605$ (lower than $0.45$ in our case).
>
> #### $R2$. **[Model Biases] “The dataset construction depends on the model performance of classification at middle steps, which may result in potential biases.”**
> $Ans$.  We thank the reviewer for their comment on the potential side-effects of supporting the data set creation with machine learning models – we acknowledge that, generally, models will always reflect the biases of their training data. This is exactly why we only keep the models’ most confident predictions through uncertainty thresholding (cf. ll. 290), or only to reduce the huge search space (cf. ll. 344) for the final manually conducted annotations. The latter is a standard technique in the literature (e.g., Simpson et. al (2018), Kuznetsov et. al (2022)) and we thus strongly believe in the high quality of our data set.
>
> #### $R3$. **[Annotation details] “Some details of the human annotation are missing. For example, the authors mention that "We develop an annotation interface based on INCEpTION (Klie et al., 2018)", but the questions for the annotation are not given.”**
> $Ans$.  We thank the reviewer for pointing out this missing detail. We will add the annotation guidelines along with the snapshots of the interface in the camera-ready version.
>
> # Questions
> #### $Q1$. **[More Powerful models] “In Section 3.2.1 Theme Prediction and Section 3.2.3 Canonical Rebuttal Identification, why don't you choose a more powerful model such as T5 or GPT for classification? Since the data consturction depends on the performance of classification, I wonder if a more powerful would be benificial.”**
>
> $Ans$. We agree with the reviewer that T5 and GPT are go-to candidates for generative tasks, and thus, we employed those for Review Description Generation (cf. ll 492) and End2End Canonical Rebuttal Generation (cf. ll 524). However, for the discriminative tasks, we focus on state-of-the-art models for scientific text analysis (Kennard et. al, (2022), Ghoshal et. al, (2022)) in-line with our goal of benchmarking strong baselines for future research. Accordingly, we test SciBERT, SPECTER, and their base variants BERT and RoBERTa (as a “more advanced BERT”). Additionally, we experiment with even finer-grained domain specialisation (for peer review texts), which we demonstrate to be effective.
>
> # Typos
> #### $T1$. **[Typo] “Typo in Line 367”**
> $Ans$. We thank the reviewer for catching this typo which we will correct in future versions of this manuscript.
>
> ### **References**
> Simpson et. al, (2018): “Finding Convincing Arguments Using Scalable Bayesian Preference Learning” (Simpson et. al, TACL 2018)
>
> Kuznetsov et. al, (2022): “Revise and Resubmit: An Intertextual Model of Text-based Collaboration in Peer Review” (Kuznetsov et. al, Computational Linguistics 2022)
>
> Kennard et. al, (2022): “DISAPERE: A Dataset for Discourse Structure in Peer Review Discussions” (Kennard et. al, NAACL 2022)
>
> Ghoshal et. al, (2022): “Peer review analyze: A novel benchmark resource for computational analysis of peer reviews” (Ghoshal et. al, PLOS One 2022)

---

### Official Review · Reviewer_7PTc · 2023-08-03

**Soundness:** 4

**Excitement:**

4: Strong: This paper deepens the understanding of some phenomenon or lowers the barriers to an existing research direction.

**Paper Topic And Main Contributions:**

This paper proposes a novel task regarding attitude and theme-guided rebuttal generation. To achieve the goal, the authors adapt the established concepts from peer review analysis and develop domain-specific models.
The contribution of this paper includes the provision of a dataset and the conduct of a thorough evaluation.


**Reasons To Accept:**

The study presents a compelling argument, supported by a comprehensive evaluation.

**Reasons To Reject:**

No significant shortcomings have been identified. However, the token adjacent to the title should be removed to align with the conventions of academic writing.



**Reproducibility:**

4: Could mostly reproduce the results, but there may be some variation because of sample variance or minor variations in their interpretation of the protocol or method.

**Reviewer Confidence:**

3: Pretty sure, but there's a chance I missed something. Although I have a good feel for this area in general, I did not carefully check the paper's details, e.g., the math, experimental design, or novelty.

---

> ### Author Rebuttal · Authors · 2023-08-28
>
> Thank you so much for your feedback and appreciating our work! We address the concerns below.
>
> # Reasons
>
> #### $R1$. **"No significant shortcomings have been identified. However, the token adjacent to the title should be removed to align with the conventions of academic writing."**
> $Ans$. We will incorporate your suggestion and update the title to “Exploring Jiu-Jitsu Argumentation for Writing Peer Review Rebuttal”.

---

### Official Review · Reviewer_5iUR · 2023-08-05

**Typos Grammar Style And Presentation Improvements:**
**Soundness:** 4

**Excitement:**

4: Strong: This paper deepens the understanding of some phenomenon or lowers the barriers to an existing research direction.

**Missing References:**



**Paper Topic And Main Contributions:**

The authors propose the task of peer-review rebuttal generation using attitude roots and themes, motivated by an argumentation style inspired by the Jiu-Jitsu combat strategy. The key contributions are
1. new task of attitude and theme-guided rebuttal generation
2. enrichment of an existing peer-review dataset with attitude roots, attitude themes, and canonical rebuttals. The authors use intermediate classification models trained on another related dataset to make annotation predictions.
3. provide baseline methods for end to end task of attitude and theme-guided rebuttal generation.

**Questions For The Authors:**

Question A: Most of the human annotations are done by one or two CS PhD students. It would benefit to provide more details of annotators, for instance, were these annotators same as the authors of the paper? If not, what was the criteria used to for recruitment besides familiarity with peer-review process? How much interaction between authors and the annotators was possible and how did it impact annotation quality?

Suggestion:
A discussion about who is intended as the user of this proposed system and what is the user expected to benefit from such a system could help strengthen the paper further.  For instance, as much as I understand, with the current framing of the paper, the intended user of the system is the author of a paper under review and suggestions of canonical rebuttals are supposed to provide ideas on how to construct a review. The system is not expected to output exact rebuttals that can be used directly, but most relevant rebuttal patterns that would require customization. I think that is a very reasonable approach and a discussion of such system use can benefit readers.

**Reasons To Accept:**

1. The task is novel (attitude and theme prediction followed by canonical rebuttal identification) and well motivated.
2. The authors provide a resource which can be used by other researchers to advance computational modeling for peer-review rebuttal generation.
3. The authors provide extensive experiments and the paper is well written.

**Reasons To Reject:**

I don't think i have any concerns.

**Reproducibility:**

4: Could mostly reproduce the results, but there may be some variation because of sample variance or minor variations in their interpretation of the protocol or method.

**Reviewer Confidence:**

3: Pretty sure, but there's a chance I missed something. Although I have a good feel for this area in general, I did not carefully check the paper's details, e.g., the math, experimental design, or novelty.

---

> ### Author Rebuttal · Authors · 2023-08-28
>
> We would like to thank the reviewer for their comprehensive feedback and for appreciating our work!
>
> # **Questions**
> #### $Q1$. **[Annotators] “Most of the human annotations are done by one or two CS PhD students. It would benefit to provide more details of annotators, for instance, were these annotators same as the authors of the paper? If not, what was the criteria used to for recruitment besides familiarity with peer-review process? How much interaction between authors and the annotators was possible and how did it impact annotation quality?”**
>
> $Ans$.  We thank the reviewer for their question and agree that more details on the selection of the annotators will enrich the discussion. Initially, we experimented with varying degrees of expertise but found that CS Masters students struggled with the task, which is why we resorted to recruiting PhD students. The final set of annotators consists of two Ph.D. students (one of whom is also author of this paper). We trained the annotators initially for 1 hour to explain the guidelines and then answered their questions when needed. We will provide a more detailed description in case of acceptance of our work.
>
> # **Suggestions**
> #### $S1$. **[UseCase] “A discussion about who is intended as the user of this proposed system and what is the user expected to benefit from such a system could help strengthen the paper further. For instance, as much as I understand, with the current framing of the paper, the intended user of the system is the author of a paper under review and suggestions of canonical rebuttals are supposed to provide ideas on how to construct a review. The system is not expected to output exact rebuttals that can be used directly, but most relevant rebuttal patterns that would require customization. I think that is a very reasonable approach and a discussion of such system use can benefit readers.”**
> $Ans$. We appreciate your suggestion on the use case and, in case of acceptance, we will add this discussion to our manuscript.

---

### Meta-Review · Area_Chair_QQPi · 2023-09-19

**Recommendation:** 4

**Metareview:**

The reviewers agreed that the novel task of attitude and theme-guided rebuttal generation based on argumentation is interesting and the proposed approach is sound. They highlighted some drawbacks in their reviews like the limited human evaluation, the lack of details about it, and the lack of deeper discussion about the potential biases on the dependence of the dataset on the model performance. The reviewers appreciated the author rebuttal which clarified most of the issues.

---

### Decision · Program_Chairs · 2023-10-07

**Decision:**

Accept-Main

**Comment:**

The reviewers agreed that the novel task of attitude and theme-guided rebuttal generation based on argumentation is interesting and the proposed approach is sound. They highlighted some drawbacks in their reviews like the limited human evaluation, the lack of details about it, and the lack of deeper discussion about the potential biases on the dependence of the dataset on the model performance. The reviewers appreciated the author rebuttal which clarified most of the issues.